# Associations between Coronavirus and Immune Response, Cardiorespiratory Fitness Rehabilitation and Physical Activity: A Brief Report

**DOI:** 10.3390/ijerph20054651

**Published:** 2023-03-06

**Authors:** Sandra Silva-Santos, António M. Monteiro, Tiago M. Barbosa, José E. Teixeira, Luís Branquinho, Ricardo Ferraz, Pedro Forte

**Affiliations:** 1Department of Sports, Higher Institute of Educational Sciences of the Douro, 4500-708 Penafiel, Portugal; 2CI-ISCE/ISCE Douro, 4500-708 Penafiel, Portugal; 3Research Center in Sports Performance, Recreation, Innovation and Technology (SPRINT-IPVC), Polytechnic Institute of Viana do Castelo, 4960-320 Viana do Castelo, Portugal; 4Department of Sport Sciences, Polytechnic Institute of Bragança, 5300-252 Bragança, Portugal; 5Research Center in Sports, Health and Human Development, CIDESD, 6201-001 Covilhã, Portugal; 6Department of Sport Sciences, Polytechnic Institute of Guarda, 6300-559 Guarda, Portugal; 7Department of Sport Sciences, University of Beira Interior, 6201-001 Covilhã, Portugal

**Keywords:** sport, immunity system, respiratory infection, recovery

## Abstract

COVID-19 has serious effects on cardiorespiratory capacity. In this sense, physical activity has been identified as beneficial in the treatment of cardiorespiratory diseases due to its anti-inflammatory and immunosuppressive benefits. To date, no study has been found on cardiorespiratory capacity and rehabilitation in patients cured after COVID-19. Thus, this brief report aims to relate the benefits of physical activity to cardiorespiratory function after COVID-19. It is important to know how different levels of physical activity can be related to the different symptoms of COVID-19. In view of this, the objectives of this brief report were to: (1) explore the theoretical associations between COVID-19 symptoms and physical activity; (2) compare the cardiorespiratory function of non-COVID-19 participants and post-COVID-19 patients; and (3) propose a physical activity program to improve the cardiorespiratory fitness of post-COVID-19 patients. Thus, we note that moderate-intensity physical activity (i.e., walking) has a greater beneficial effect on immune function, whereas vigorous activity (i.e., marathon running) tends to temporarily reduce immune function through an imbalance of cytokine types I and II in the hours and days after exercise. However, there is no consensus in the literature in this regard, since other investigations suggest that high-intensity training can also be beneficial, not causing clinically relevant immunosuppression. Physical activity has been shown to be beneficial in improving the clinical conditions most frequently associated with severe COVID-19. Thus, it is possible to infer that physically active individuals seem to be less exposed to the dangers of severe COVID-19 compared to non-active individuals through the benefits of physical activity in strengthening the immune system and fighting infections. The current study demonstrates that physical activity appears to be beneficial in improving the clinical conditions most often associated with severe COVID-19.

## 1. Introduction

The SARS-CoV-2 (also known as COVID-19), is an acute respiratory disease spread worldwide, leading to one of the worst pandemics in recent times [1,2,3]. COVID-19 has some significant effects on the cardiorespiratory system. COVID-19 patients may present the acute respiratory distress syndrome, a respiratory failure characterized by lung inflammation. Some of its effects include shortness of breath, rapid breathing, blue skin coloration and blood coagulopathy [4].

Since this is a new disease, research is needed to better understand the acute and chronic effects, especially on the cardiorespiratory system, including lung functions. There are some studies evaluating post mortem pulmonary anatomy in patients with and without COVID-19 [1,2,3,5].

One study reported that after death, two distinctive immunopathological patterns were detected: one presented high local expression of interferon stimulated genes (ISG) and cytokines, high viral loads and limited pulmonary damage; the other pattern showed severely damaged lungs, low ISGs (ISG), low viral loads and abundant infiltrating activated CD8+ T cells and macrophages [2]. ISGs, cytokines and CD8+ T are typically responsible for immune responses, pathogenesis resistance and the control, identification and destruction of cells and degrade and digest viruses [1,2,3].

In another investigation, post-mortem patients with COVID-19 presented higher severe lung vascular (endothelial) injury with high a prevalence of T-cells, thrombosis and microangiopathy [3]. Some postmortem patients had expressive alveolar damage and massive capillary congestion by microthrombi [1]. Moreover, COVID-19 has been associated with myocarditis, cardiac arrest, and acute heart failure. In summary, COVID-19 disease is related to impairment of lung activity, cardiorespiratory function, and the main cause of death is respiratory failure [1,2,3].

Discharged patients after COVID-19 infection still present residual pulmonary abnormalities [6]. The literature reports that previous coronavirus variants impaired the subjects’ lung function and exercise capacity for months or years, and this topic still needs clarification [7]. These data gain emphasis since the potential impact of abnormalities in lung function with reduced exercise capacity has been reported [8]. Cardiac rehabilitation helps to improve functional capacity and health-related quality of life by improving cardiorespiratory fitness [9]. So far numerous benefits of pulmonary rehabilitation have been reported, such as improvements in dyspnea, functional capacity, and exercise performance, lung function and quality of life [10,11,12].

However, studies on the rehabilitation of patients with pulmonary embolism are lacking due to the lack of consensus on the eligibility criteria for rehabilitation [13,14,15]. COVID-19 patients can be symptomatic (with severe or non-severe inflammatory damage) or asymptomatic (without inflammatory damage). However, little is known about asymptomatic patients [16]. That said, the injuries associated with the disease may also differ from patient to patient.

Symptomatic patients can have moderate to severe lungs illness and the cardiorespiratory effects might be different depending on the symptom’s prevalence [17]. At least one study recommended the following treatments for post-COVID-19 patients: respiratory physio-kinesiotherapy and postural drainage of the lungs; mechanical pulmonary ventilation as rehabilitation therapy (with or without drugs); oxygen therapy; inhalation therapies with mineral waters; physical activity and psychological support [18].

Low intensity in-bed exercises in persons with physical impairment or incapacity induced low-level muscular activity. However, it is possible to obtain increased cardiac output and physiological cardiorespiratory response [19]. Moreover, exercise-based cardiac rehabilitation with moderate and moderate-to-vigorous intensities were associated with moderate improvements in absolute peak oxygen uptake (VO_2peak_) [9].

Data on pulmonary embolism (a COVID-19 symptom that may lead to alveolar injury) rehabilitation are scant [9]. During the rehabilitation, the non-existence of severe events such as bleeding and rehospitalization for venous thromboembolism define the rehabilitation as safe [15]. Regarding physical activity and physical exercise, anti-inflammatory and immunosuppressive effects are based on cytokine and T-cell production [20,21]. The cytokine (anti-inflammatory) response is higher with high-intensity physical exercise [22]; moreover, low to moderate physical exercise increases the blood concentration of T-cells [20]. These cells are responsible for the immunosuppressive response.

These biochemical markers contribute to the anti-inflammatory response against COVID-19 [2,3,17]. Therefore, physical activity may become an important key factor to mediate the anti-inflammatory response to the COVID-19 illness.

Based on their physical activity levels, people may be classified as sedentary or active. However, the recommendations are based on physical activity intensity and accumulated volume [23]. Adults (i.e., 18–64 years old) should spend at least 150–300 min/week engaged in moderate aerobic physical activity or 75–150 min/week of vigorous aerobic physical activity and engage in strength training twice a week [24]. For additional gains, the adults may have to engage in more than 300 min/week of moderate aerobic physical activity or more than 150 min/week of aerobic vigorous physical activity [23]. Older people (over 65 years old) can follow the adult guidelines. However, they must vary with multicomponent activities for 3 days/week [23].

As mentioned above, physical activity plays an important role in anti-inflammatory and immunosuppressive effects [9]. COVID-19 has serious severe effects on cardiorespiratory capacity [1,2,3]. Conversely, physical activity has a positive effect on cardiorespiratory illness treatments [23]. The literature reported that multiple sessions between 1 and 10 min for persons with limited capacities are recommended [25]. Moreover, the aerobic conditioning segment may last between 20 min and 60 min [25]. So far, no study has been carried out on the cardiorespiratory capacity and rehabilitation in post-COVID-19 healed participants.

Therefore, this project may contribute by relating the benefits of physical activity to post-COVID-19 cardiorespiratory function. It might be expected that high levels of physical activity may be related to fewer symptoms or with lower severity in people infected with COVID-19 due to the anti-inflammatory and immunosuppressive effects physical activity and exercise. The literature about assessing physical activity levels in post-COVID-19 healed persons and their cardiorespiratory fitness is scarce. It is important to know how different physical activity levels are related to the different symptoms of COVID-19 [26,27]. Moreover, there is a need to understand how physical activity can help to improve post-COVID-19 healed persons’ cardiorespiratory fitness to similar levels of non-infected people.

As far the authors know, there is no study assessing the physical activity of post-COVID-19 healed persons. There are objective and subjective methods with different sensors or instruments to assess physical activity such as: heart rate monitors; the perceived exertion assessment scale; units of distance traveled per unit of time; questionnaires; accelerometry; and pedometers [23,25,28]. Given this, research questions such as: “How can physical activity levels be related to the symptoms of patients with COVID-19?”, “How can people who are cured post-COVID-19 differ in cardiorespiratory fitness compared to non- COVID-19?” and “Is it possible to improve cardiorespiratory fitness through a physical activity program to levels similar to uninfected people?” [26,27,29], still remain without evidence-based answers, and we hope that this study will help to resolve some questions. Physical activity levels may play an important role in anti-inflammatory and immunosuppressive effects against COVID-19 [9] and post-disease cardiorespiratory rehabilitation [1,2,3,9]. It is important to note that the current evidence collected is not based on experimental or quasi-experimental research. Due to pandemic restrictions, most surveys made use of questionnaires and similar survey tools [30,31,32].

It is essential to assess the levels of physical activity of people cured after COVID-19. Cardiorespiratory fitness in cured and uninfected post-COVID-19 participants should be measured in real-time online sessions with simple physical fitness tests. Finally, it is worth clarifying whether increased physical activity enables a rapid recovery of post-COVID-19 patients, notably cardiorespiratory fitness.

The objectives of this brief report were to: (1) explore the theoretical associations between COVID-19 symptoms and physical activity; (2) compare cardiorespiratory function between non-COVID-19 participants and post-COVID-19 patients; and (3) propose a physical activity program to improve the cardiorespiratory fitness of post-COVID-19 patients. It was hypothesized that the literature would reveal: (1) associations between physical activity and the number of symptoms and cardiorespiratory capacity in post-COVID-19 patients; (2) non-COVID-19 patients have better cardiorespiratory fitness compared to post-COVID-19 patients; and (3) a physical activity program can improve the cardiorespiratory capacity of post-COVID-19 cured people to levels similar to non-infected people.

## 2. Materials and Methods

### 2.1. Literature Search Strategy and Selection Criteria

The literature search was conducted using a Boolean operator on four databases between October and December 2022: PubMed/Medline, Web of Science, Google Scholar and SportDiscus using the primary keywords “COVID-19”, “SARS-CoV-2” and “immune response” associated with the secondary keywords: “cardiorespiratory fitness”, “rehabilitation” and “physical activity”. The selection was made by an independent author (P.F.) and verified by a second author (L.B.). Discrepancies between authors in the selection of studies were resolved by a third reviewer (R.F.). The criteria were adopted based on the PRISMA guidelines that recommend double review [33]. The literature search was limited to peer-reviewed articles and the authors did not prioritize authors or journals.

### 2.2. Selection Criteria

The inclusion criteria were for articles which: (1) contained data related to the impact of COVID-19 on cardiorespiratory fitness in all types of patients and ages; (2) contained data related to cardiorespiratory rehabilitation for physical activity after COVID-19 in all types of patients and ages; (3) contained data related to cardiorespiratory rehabilitation for physical activity in other types of patients without COVID-19; (4) were published in English. Studies were excluded if they: (1) did not include relevant data on cardiorespiratory rehabilitation for physical activity; or (2) were conference abstracts. After this procedure, 44 article were considered for the results (Figure 1).

## 3. Results

### 3.1. The Associations between COVID-19 Symptoms and Physical Activity

Each of the new variants of COVID-19 challenges human innate immunity to respond to virus infection [34]. It is now known that, like other coronaviruses, COVID-19 causes an infection in host cells via a spike protein that binds to angiotensin-converting enzyme 2 receptors found on many human cells (e.g., such as lung epithelium) [35]. In this regard, previous evidence seems to indicate that physical activity has the potential to reduce the severity of COVID-19 infections. This fact seems to be related to what happens in the lungs when exposed to an infection via the typology of the immune response. The conflict between the virus and immune cells creates inflammation and can lead to lung tissue damage, which will have consequences on gas exchange and ultimately require mechanical ventilators for treatment [17]. Positive effects on these physiological mechanisms have been previously reported [36,37,38] as a consequence of physical activity, as muscles produce compounds that improve the functioning of the immune system and reduce inflammation. Although the reported studies have not been performed in COVID-19 patients, the literature shows the benefits of exercise in strengthening immunity against viral respiratory infections [36,37,39,40,41,42]. In this regard, some evidence seems to suggest that moderate-intensity physical activity (i.e., walking) has a greater beneficial effect on these physiological mechanisms, whereas vigorous activity (i.e., marathon running) tends to temporarily reduce immune function through of an imbalance of type I and II cytokines in the hours and days following exercise [43,44]. However, there is no consensus in the literature in this regard, given that other investigations have suggested that high-intensity training can also be beneficial, not causing clinically relevant immunosuppression [37,43]. The lack of consensus on the subject reinforces the idea that more studies are needed to verify the most appropriate exercise intensity to boost the immune system, but does not call into question the effectiveness of physical exercise in combating COVID-19.

Other studies have indicated that the most fragile populations (i.e., the elderly, immunocompromised patients or patients with multiple comorbidities) are more exposed to severe COVID-19 [45], and that diabetes, hypertension and cardiovascular diseases were reported as the most frequent pathologies among patients infected with COVID-19 who needed to be hospitalized [46]. Also in this sense, the beneficial potential of physical activity must have been taken into account, given that previous studies show its effectiveness for the prevention and treatment of the pathologies reported above [44,47]. Although health authorities recommend the practice of physical exercise, there do not seem to be any joint efforts with government entities to promote active and healthy lifestyles before exposure to the virus, in order to reduce the severity of the disease worldwide [48]. In fact, exercise has an acute effect on the functioning of the immune system and fighting inflammation [38], which can be beneficial in fighting serious viral infections such as COVID-19.

In addition, as a consequence of the COVID-19 pandemic, several factors (i.e., lockdowns, unemployment, preventive measures enacted by governments) have potentiated symptoms of stress and depression among the population [49,50,51,52], and here, too, physical exercise may have a key role [53]. Previous evidence [54] reported that individuals without regular physical exercise habits are more exposed to psychological stress, which potentiates imbalances in the levels of cortisol and other hormones with direct consequences for the immune system and the response to inflammation related to COVID-19. Recent data indicate that worldwide, 32% of women and 23% of men are at risk of being infected due to their inactivity habits [23] and this reinforces the need for the population to be made aware of the benefit of exercise in reducing the risk of infection.

Thus, there seems to be evidence on the strong association between physical activity and the main symptoms caused by COVID-19 [55,56]. Physical activity has proven to be beneficial in improving the clinical conditions most often associated with severe COVID-19 [57]. Thus, it is possible to infer that physically active individuals appear to be less exposed to the dangers of severe COVID-19 compared to non-active individuals through the benefits of physical activity in strengthening the immune system and fighting infections.

### 3.2. COVID-19, Cardiorespiratory Fitness and Immune Response

Several studies have shown that individuals with high levels of cardiorespiratory fitness have a reduced risk of severe illness and death from COVID-19 [58,59]. This may be because exercise can improve immune function, making it easier for the body to fight off infections [60,61,62]. The COVID-19 confinement is well documented in the literature to delay the normal development of VO_2_ max in teenagers, and alternative ways to minimize this delay must be created [63,64]. Strategies to address this alarming decrease are required. However, COVID-19 (infection or prophylactic isolation) has impaired not only cardiorespiratory fitness in youths, but also general physical fitness [63,65]. This delay in teenagers suggests that adults’ cardiorespiratory fitness may be diminished. Following that, engaging in sports, physical exercise, and engaging in a high degree of physical activity may help to lessen the inflammatory response. The greater one’s physical fitness, the less likely one is to acquire “cytokine storm syndrome,” a group of connected medical diseases in which the immune system produces too many inflammatory signals, sometimes leading to organ failure and death. Furthermore, “high risk patients” are more likely to develop “cytokine storm syndrome” [66].

There are different methods and protocols to assess cardiorespiratory capacity. During the world confinement, online real-time evaluation sessions are possible to do, avoiding face-to-face contacts and minimizing the risks of infection.

Simple tests such as a six-minute walk or two-minute step test are possible to perform during online synchro sessions. However, patients confined to their houses may have limited space, and the two-minute step-test is an alternative to assess cardiorespiratory capacity [67]. This test was firstly designed for the elderly. However, the step test is a good alternative to assess adults’ cardiorespiratory capacity under rehabilitation [68]. The evaluator schedules sessions with volunteers to perform the two-minute step test to assess the participants’ cardiorespiratory fitness. This procedure is carried out for the total sample, monitoring both post-infected healed COVID-19 and non-infected participants.

### 3.3. Cardiorespiratory Fitness Rehabilitation and Physical Activity

After assessing post-COVID-19 cardiorespiratory capacity, the sedentary participants is encouraged to apply for a physical activity program. It is expected that, after 9 months, the habits of physical activity are preserved [69], and so the gains related to PA programs are maintained and improved. However, cardiorespiratory capacity may be evaluated every three months [70], with a total of four evaluations during the intervention. However, some cases require cardiorespiratory rehabilitation prior to starting physical activity programs [71]. Therefore, it is important to establish methods, techniques and procedures to improve cardiorespiratory fitness before starting the daily life activities.

A commercial hand-held resistance device can be used to train respiratory muscles. Three sets of ten breaths each at 60 percent of the individual’s maximum expiratory mouth pressure, with a one-minute break between sets. Respiratory muscle training was allowed. In addition, three sets of ten vigorous coughing exercises are included in the rehabilitation program. In the supine posture, each participant performed 30 maximal voluntary diaphragmatic contractions while holding a medium weight against the anterior abdominal wall to oppose diaphragmatic descent [72]. Stretching activities are useful for stretching the respiratory muscles under the supervision of a rehabilitation specialist. To straighten the lumbar curvature, patients were placed in a supine or lateral position with knees bent. Patients were instructed to move their arms in the following directions: flexion, horizontal extension, abduction, and external rotation. Subjects were schooled in pursed lip breathing and cough training at home, with 30 sets required each day.

The physical activity program objectives may be twofold: (i) aerobic-based exercises; and (ii) strength-based exercises. Therefore, multimodal training is recommended [73]. The physical activity program guidelines to improve post-COVID-19 patients’ cardiorespiratory fitness may be presented in the literature for evidence-based practices. However, special attentions to oxygen drops should be considered (SpO_2_ < 88%; Borg > 6; reaching submaximal heart rate) [74].

To cure and improve cardiorespiratory fitness in post-COVID-19 patients, physical activity programs can follow the following guidelines:

Strength training can be performed with three sets of 20 repetitions of the maximum load tolerated. The intensity of monitored resistance training sessions should be adjusted between sessions. Respiratory physiotherapy should be included. For aerobic exercises, intensity is possible from the initial six-minute walk test. Typical aerobic exercises for COVID-19 patients are walking and cycling [74].

## 4. Conclusions

The current study demonstrates that physical activity has proven to be beneficial in improving the clinical conditions most often associated with severe COVID-19. One of the biggest novelties of this study is the fact that it seeks to provide practical recommendations on physical activity programs to improve post-COVID-19 patients’ cardiorespiratory fitness.

Increasing physical activity enables a speedy recovery of post-COVID-19 patients, notably their cardiorespiratory fitness. In fact, exercise has an acute effect on the functioning of the immune system and fighting inflammation, which can be beneficial in fighting serious viral infections such as COVID-19. It is possible to infer that physically active individuals appear to be less exposed to the dangers of severe COVID-19 compared to non-active individuals, through the benefits of physical activity in strengthening the immune system and fighting infections. To heal and improve cardiorespiratory fitness in post-COVID-19 patients, the physical activity program may follow the present guidelines such as: strength training can be performed with three sets of 20 repetitions of the maximum tolerated load. The intensity of the monitored endurance training sessions should be adjusted between sessions. Respiratory physiotherapy should be included. For the aerobic exercises, intensity may be derived from initial 6 min walking test. The typical aerobic-based exercises for COVID-19 patients are walking and bicycle.

However, this study is not without its limitations, and this is largely due to the fact that the current evidence collected is not based on experimental or quasi-experimental research. Due to the constraints of the pandemic, most research resorted to surveys, questionnaires and similar research instruments, and for these reasons there is an urgent need to deepen research on the subject. In addition, it would be important for further investigations to focus on recommending individualized physical activity programs based on different conditions that may have resulted from exposure to COVID-19, as this factor could be decisive in enhancing the recovery of more patients.

## Figures and Tables

**Figure 1 ijerph-20-04651-f001:**
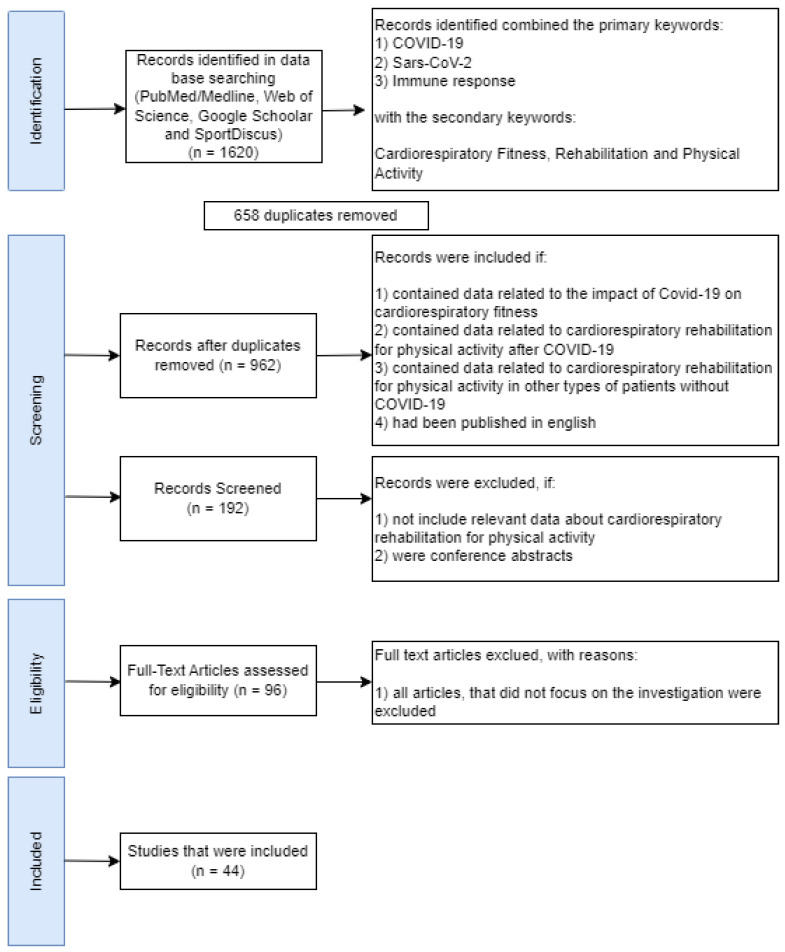
Flow diagram adapted from PRISMA 2009.

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
