# Peer review of "Associations between Coronavirus and Immune Response, Cardiorespiratory Fitness Rehabilitation and Physical Activity: A Brief Report"

_ijerph, 2023, doi:10.3390/ijerph20054651_

Round 1

Reviewer 1 Report

I appreciate the opportunity to review this study.

This is a brief report that described associations between coronavirus and immune response, cardiorespiratory fitness rehabilitation, and physical activity and is a valuable short review on the field of indicating the physical activity effects on COVID-19. However, the methodology and research content of this brief report are insufficient to reveal the association proposed by the authors in the Title. I believe that significant modifications are needed as follows.

Overall comment

ž  Despite that the authors indicated that the effects of COVID-19 vary with age and chronic disease status, the participants targeted in this report are undefined and to whom the purpose of this brief report is intended is unclear; therefore, the results and conclusions drawn are also unclear. Additionally, the references cited are slightly outdated, and several recent studies have provided references to clarify the report’s objectives. It would be more meaningful to recheck those references and update the information.

ž  "Covid-19" needs to be capitalized.

 Individual comments

Abstract

Since the evaluation of cardiorespiratory fitness appears abruptly, please modify the structure considering the text flow. Ultimately, it would be easier for readers to understand if the conclusion of this brief study is presented.

Materials and methods

I believe it is necessary to include the additional information on inclusion criteria, such as targeted participant’s age and the range of publication date.

I believe that following the dual review process and other PRISMA guideline items will provide the authors with a more rigorous and reliable literature evaluation and increase the impact and usefulness of the results.

If this study is registered with PROSPERO, please provide the number.

Results

It would be easier to understand the study results if specific examples of the 39 selected references were provided.

Apparently, there are many areas that the association between physical activity andCOVID-19 has been speculated using previous studies before the COVID-19 expansion.

Since several recent studies (e.g., Front Physiol. 2022. doi: 10.3389/fphys.2022.1030568; IJERPH 2022.doi: 10.3390/ijerph19159025; IJERPH 2022. doi:10.3390/ijerph192114108) have shown associations, please review the most recent information and reconsider the review to update it with studies that present the actual association with COVID-19.

Author Response

Reviewer 1

Overall comment

Despite that the authors indicated that the effects of COVID-19 vary with age and chronic disease status, the participants targeted in this report are undefined and to whom the purpose of this brief report is intended is unclear; therefore, the results and conclusions drawn are also unclear. Additionally, the references cited are slightly outdated, and several recent studies have provided references to clarify the report’s objectives. It would be more meaningful to recheck those references and update the information.

R: Dear reviewer, first of all thank you for agreeing to review our article. Your comments were important to improve the quality of the article and we are grateful for that.

Below you can find our answers in detail.

"Covid-19" needs to be capitalized.

R: Dear reviewer, thanks for the suggestion, the changes have been made.

Individual comments

Abstract

Since the evaluation of cardiorespiratory fitness appears abruptly, please modify the structure considering the text flow. Ultimately, it would be easier for readers to understand if the conclusion of this brief study is presented.

 R: Dear reviewer, thank you for your comments. the content of the abstract has been reorganized.

Materials and methods

I believe it is necessary to include the additional information on inclusion criteria, such as targeted participant’s age and the range of publication date.

R: Dear reviewer, thank you for your comments. the content of the abstract has been reorganized, and the requested information entered.

I believe that following the dual review process and other PRISMA guideline items will provide the authors with a more rigorous and reliable literature evaluation and increase the impact and usefulness of the results.

If this study is registered with PROSPERO, please provide the number.

R: Dear reviewer, thank you for your comments. The study in question is a Brief report and for these reasons it was registered in PROSPERO.

Results

It would be easier to understand the study results if specific examples of the 39 selected references were provided.

Apparently, there are many areas that the association between physical activity andCOVID-19 has been speculated using previous studies before the COVID-19 expansion.

Since several recent studies (e.g., Front Physiol. 2022. doi: 10.3389/fphys.2022.1030568; IJERPH 2022.doi: 10.3390/ijerph19159025; IJERPH 2022. doi:10.3390/ijerph192114108) have shown associations, please review the most recent information and reconsider the review to update it with studies that present the actual association with COVID-19.

R: Dear reviewer, thank you very much for your comments. The results of the studies included in the results were not detailed in a table because this article is a brief report that intends to raise the study problem. This study is the first of a series of studies that is being carried out by this work team on the topic and in that sense one of the studies includes a systematic review. We hope you understand our reasons-

The studies presented prior to the Coronavirus contain relevant information on the importance of cardiorespiratory rehabilitation for physical activity in another type of patient, but which contain useful information when aggregated with information from more recent articles. By mistake, this information was not included in the inclusion criteria but was inserted.

The studies you suggest were included in the manuscript.

Reviewer 2 Report

The paper is good and sounds , but there are some minor comments 

1- Add future work research on this subject 

2- The introduction's paragraph are too big I suggest splitting them into small ones

3- The  Regernce number 45 is old , I suggest replacing it

4- The novelty should be added in a single paragraph in the introduction

5- The differences between this paper and the other papers in the literature should be well defined 

6- The English should be revised, there are some grammatical mistakes and flows 

Author Response

Reviewer 2

Dear reviewer, first of all thank you for agreeing to review our article. Your comments were important to improve the quality of the article and we are grateful for that.

Below you can find our answers in detail where we try to meet all your requests.

The paper is good and sounds, but there are some minor comments.

1- Add future work research on this subject 

R: Dear reviewer, thank you for your comment, potential suggestions for future investigations have been added in the conclusion.

2- The introduction's paragraph are too big I suggest splitting them into small ones

R: Dear reviewer, we totally agree with your observation and accordingly, the paragraphs were divided.

3- The Reference number 45 is old, I suggest replacing it

R: Dear reviewer, based on your recommendations the reference has been changed to a newer one.

4- The novelty should be added in a single paragraph in the introduction

R: Dear reviewer, thank you for your comment, we reinforce the novelty of the study in the introduction.

5- The differences between this paper and the other papers in the literature should be well defined

R: Dear reviewer, thank you for your comments, a statement regarding the novelty of the study has been added in the conclusion.

6- The English should be revised, there are some grammatical mistakes and flows 

R: Dear reviewer, we try to improve the document according to your indications.

Reviewer 3 Report

Comments to the Author

The authors of this article did an admirable job on an important topic, aimed to explore the associations between COVID-19 and the immune response, cardiorespiratory fitness, and physical activity level. This paper is well-organized, pertinent, and may add to the literature base of an important. However, there are several points that require further clarity;

1- Page 2, Lines 58-59: Please revise ‘’post-mortom’’ as post-mortem

2- Page 2, Lines 66-68: In this paragraph, it is necessary to indicate how long the abnormal respiratory functions (pulmonary function and respiratory muscle strength) persist in the normal and athlete populations. In addition, the relationship between abnormal pulmonary functions and decreased exercise capacity can be partially mentioned.

3- Page 2, Lines 68-70: The cardiac and pulmonary systems are combined systems. Recent studies have shown that some pulmonary rehabilitation practices (IMT etc.) have several benefits for patients recovering from COVID-19. Please mention the benefits of pulmonary rehabilitation with references.

4- Page 3, Lines 120-121: Following this sentence, the most common symptoms seen in the normal population and the most common symptoms seen in the population of athletes with high physical activity levels can be stated or some references can be included here for your statement. See the discussion section second paragraph. This can be a good reference for your article.

https://doi.org/10.1016/j.resp.2022.103983

5- Page 3, Line 134: Studies showing cardiac, pulmonary function or symptomatic abnormalities by physical activity level after COVID-19 should be cited in this paragraph or in section 3.1 below (if any). or the differences in outcomes between athletes and the general population better be shown. See the studies

https://doi.org/10.1016/j.resp.2022.103983

http://dx.doi.org/10.1136/bjsports-2021-104080

http://dx.doi.org/10.1136/bjsports-2020-102789

5- Page 6, Lines 229: This section needs further development.

6- Page 7, Lines 253: It would also be better if this section also provides some more comprehensive information about the benefits of IMT or similar applications for COVID-19 patients.

GENERAL COMMENTS:

1. The manuscript requires language improvement.

2. The topic is important but especially the introduction and discussion sections should be improved significantly. Literature review is nonadequacy.

3. Abstract should be re-edited after changes made in the article.

Author Response

Reviewer 3

Dear reviewer, first of all thank you for agreeing to review our article. Your comments were important to improve the quality of the article and we are grateful for that.

Below you can find our answers in detail where we try to meet all your requests.

1- Page 2, Lines 58-59: Please revise ‘’post-mortom’’ as post-mortem

R: Dear reviewer, the word has been changed in accordance with your recommendations.

2- Page 2, Lines 66-68: In this paragraph, it is necessary to indicate how long the abnormal respiratory functions (pulmonary function and respiratory muscle strength) persist in the normal and athlete populations. In addition, the relationship between abnormal pulmonary functions and decreased exercise capacity can be partially mentioned.

R: Dear reviewer, the paragraph has been changed according to your recommendations.

3- Page 2, Lines 68-70: The cardiac and pulmonary systems are combined systems. Recent studies have shown that some pulmonary rehabilitation practices (IMT etc.) have several benefits for patients recovering from COVID-19. Please mention the benefits of pulmonary rehabilitation with references.

R: Dear reviewer thanks for your comment pulmonary rehabilitation benefits have been added.

4- Page 3, Lines 120-121: Following this sentence, the most common symptoms seen in the normal population and the most common symptoms seen in the population of athletes with high physical activity levels can be stated or some references can be included here for your statement. See the discussion section second paragraph. This can be a good reference for your article.

https://doi.org/10.1016/j.resp.2022.103983

R: Dear Reviewer, References have been added as per your recommendations.

5- Page 3, Line 134: Studies showing cardiac, pulmonary function or symptomatic abnormalities by physical activity level after COVID-19 should be cited in this paragraph or in section 3.1 below (if any). or the differences in outcomes between athletes and the general population better be shown. See the studies

https://doi.org/10.1016/j.resp.2022.103983

http://dx.doi.org/10.1136/bjsports-2021-104080

http://dx.doi.org/10.1136/bjsports-2020-102789

R: Dear Reviewer, References have been added as per your recommendations.

5- Page 6, Lines 229: This section needs further development.

R: Dear Reviewer, the point was developed and rearranged.

6- Page 7, Lines 253: It would also be better if this section also provides some more comprehensive information about the benefits of IMT or similar applications for COVID-19 patients.

R: Dear reviewer, thank you for your comments. The main objective of this article is fundamental to raise the problem of study. This study is part of a set of studies that is being carried out and for this reason the inclusion of more information in this regard may condition a systematic and exhaustive review that we are carrying out. We hope you understand our reasons.

GENERAL COMMENTS:

  1. The manuscript requires language improvement.
  2. The topic is important but especially the introduction and discussion sections should be improved significantly. Literature review is nonadequacy.

R: Dear reviewer, try to adjust the manuscript according to your recommendations.